# Influence of Antimicrobial Resistance on the Course of Symptoms in Female Patients Treated for Uncomplicated Cystitis Caused by *Escherichia coli*

**DOI:** 10.3390/antibiotics11020188

**Published:** 2022-01-31

**Authors:** Marie Soees Waldorff, Lars Bjerrum, Anne Holm, Volkert Siersma, Christine Bang, Carl Llor, Gloria Cordoba

**Affiliations:** 1Research Unit for General Practice and Section of General Practice, Department of Public Health, University of Copenhagen, 1350 Copenhagen, Denmark; lbjerrum@sund.ku.dk (L.B.); anneholm@sund.ku.dk (A.H.); tjs657@sund.ku.dk (V.S.); cwb@econ.ku.dk (C.B.); gloriac@sund.ku.dk (G.C.); 2University Institute in Primary Care Research Jordi Gol, Via Roma Health Centre, 08007 Barcelona, Spain; cllor@health.sdu.dk; 3Department of Public Health, General Practice, University of Southern Denmark, 5000 Odense, Denmark

**Keywords:** drug resistance, microbial, urinary tract infections, uropathogenic *Escherichia coli*, primary health care, patient-reported outcome measures, treatment outcome, Denmark

## Abstract

Background: Resistance to the prescribed antibiotic causes a longer duration of symptoms in patients with urinary tract infection. Yet, a study found that patients infected with trimethoprim-resistant *Escherichia coli* (*E. coli*) had a prolonged duration of symptoms even if treated with an antibiotic to which the strain was susceptible. The purpose of this study was to attempt to reproduce this finding in a different cohort. Methods: We analyzed data from two studies from general practice in the Capital Region of Denmark including patients from 2014 to 2016. The primary outcome was the severity of frequency and dysuria. The secondary outcome was the number of days until symptoms had disappeared. Results: We included 180 women treated for uncomplicated cystitis caused by *E. coli*. We found that 16.11% (*n* = 29) of the *E. coli* strains were resistant to all of the three selected antibiotics (ampicillin, sulfamethizole and trimethoprim). There was no significant difference in severity or time until the symptoms had disappeared between women infected with resistant or susceptible *E. coli*. Conclusions: Strains of *E. coli* resistant to ampicillin, sulfamethizole and trimethoprim causing uncomplicated cystitis do not result in more severe symptoms or a longer symptom duration if treated with an antibiotic to which they are susceptible.

## 1. Introduction

Urinary tract infection (UTI) is one of the most frequent infections encountered in general practice and about one in three women experience a UTI at least once in their life time [1,2].

The increase in antibiotic resistance is a threat to human and animal health worldwide and the excessive use of antibiotics in general practice continues to be an unfortunate driver towards the development of antibiotic resistance, both in society and in the individual patient [3,4,5]. For example, the high use of quinolones for uncomplicated UTIs in the early 2000s resulted in a gradual increase in uropathogens resistant to these antibiotics. As a result, these antimicrobials must now be used more judiciously [6].

*E. coli* is the most frequent uropathogenic bacteria, and in Denmark, pivmecillinam is the antibiotic most commonly prescribed for UTI [7,8]. In Denmark, the prevalence of *E. coli* resistant to pivmecillinam is 5% in samples from primary care [9]. In the United Kingdom, trimethoprim is one of the first-line antibiotics recommended for UTI and the resistance rate against trimethoprim for *E. coli* is 14–17% [10,11,12]. Resistance rates for other antibiotics used in primary health care in Denmark to treat UTIs are ampicillin (36%), sulfamethizole (27%), trimethoprim (21%), nitrofurantoin (0.5%) [9].

Studies from general practice have shown that patients with UTI caused by uropathogen bacterial strains resistant to one or more antibiotics have more severe symptoms and a longer symptom duration than patients with UTI caused by susceptible strains [10,11,13]. Most studies have investigated the influence of uropathogens that are resistant to the prescribed antibiotic. However, a study from the United Kingdom also found a longer symptom duration if the uropathogenic bacteria, *Escherichia coli (E. coli*), was resistant to trimethoprim in spite of being susceptible to the prescribed antibiotic [11].

The different antibiotic recommendations for UTI in Denmark and the United Kingdom, as well as the different resistance patterns, make it relevant to investigate if the findings from the study from the United Kingdom are applicable to the Danish population.

Hence, the aim of this analysis was to investigate the influence of the antimicrobial resistance (AMR) pattern on the course of symptoms in women with uncomplicated cystitis caused by *E. coli* treated with an antibiotic to which the bacteria is susceptible. We chose to focus on women with uncomplicated cystitis in order to have a more homogenous group.

## 2. Results

In total, 59 general practices included 851 patients for the two studies. Figure 1 shows a flowchart of the patients included and excluded from the analyses in this paper. We included 180 women with uncomplicated cystitis caused by *E. coli* and treated with pivmecillinam, sulfamethizole, trimethoprim or nitrofurantoin on the day after consultation for this analysis. Out of the 671 excluded patients, 436 were not infected with *E. coli*, 79 had a complicated UTI or upper UTI, 127 patients were not prescribed antibiotics as stated in our inclusion criteria and 29 had not returned their symptom diaries.

Table 1 shows the baseline data. The age of the women ranged from 18 to 90 years, with 59 women being 65 years of age or older. The majority of patients (*n* = 114, 63.33%) were prescribed pivmecillinam. Resistance rates were highest for ampicillin (*n* = 64, 35.56%), followed by sulfamethizole (*n* = 48, 26.67%), and trimethoprim (*n* = 43, 23.89%). Twenty-nine (16.11%) of the urine samples showed growth of *E. coli* that were resistant to all of the three antibiotics (i.e., ampicillin, sulfamethizole and trimethoprim). The baseline characteristics were similar for women with missing data in their symptom diaries (data not shown).

The mean symptom severity score of frequency and dysuria was not influenced by the AMR pattern (see Figure 2 and Figure 3). The difference in severity score was highest on day one, that is, before the initiation of antibiotic treatment. In patients with *E. coli* resistant to all three antibiotics (ampicillin, sulfamethizole and trimethoprim), the frequency severity score at day one was 7.0 (95% CI: 6.0–7.9), 7.7 (95% CI: 6.9–8.4) in patients with *E. coli* resistant to one or two antibiotics and 7.5 (95% CI: 7.0–8.0) in patients with *E. coli* sensitive to all three antibiotics. The difference was not significant. On day five, the frequency severity scores were below two, irrespective of the AMR pattern. In patients with *E. coli* resistant to all three antibiotics (ampicillin, sulfamethizole and trimethoprim), the dysuria severity score at day one was 4.0 (95% CI: 3.3–4.7), 4.9 (95% CI: 4.4–5.5) in patients with *E. coli* resistant to one or two antibiotics and 4.4 (95% CI: 4.1–4.8) in patients with *E. coli* sensitive to all three antibiotics. At day five, the dysuria severity scores were below one, irrespective of the AMR pattern.

Table 2 shows the influence of the AMR pattern on the mean number of days until symptoms had disappeared. In patients with *E. coli* resistant to all three antibiotics (ampicillin, sulfamethizole and trimethoprim) the mean number of days until overall cure was non-significantly longer (0.5 days, *p* > 0.05) than in patients with *E. coli* susceptible to all three antibiotics.

## 3. Discussion

### 3.1. Principal Findings

This analysis shows that the antimicrobial resistance pattern has no influence on the course of symptoms (frequency and dysuria) in women with uncomplicated cystitis caused by *E. coli* treated with an antibiotic to which the strain is susceptible.

### 3.2. Strengths and Weaknesses

The most important strength of this analysis is that we used a validated questionnaire to evaluate the symptom scores (Holm–Cordoba UTI Score). It ensures that the most relevant symptoms in patients with uncomplicated cystitis were accurately evaluated. Another strength is the homogeneous study population of women with uncomplicated cystitis caused by *E. coli* that were prescribed an antibiotic to which the strain was susceptible. With the use of narrow inclusion criteria for the analysis, symptom scores were not compared across a broad spectrum of the disease, which could have biased the thresholds for severity and the number of days until symptoms had disappeared. Furthermore, the population is representative of the source population in that there were no differences in the baseline characteristics between patients returning the symptom diary and those who did not do so.

The relatively small sample of patients could have hidden real differences between the groups. However, the analysis showed only very small differences (less than one day) across groups and the relatively small sample size does not outweigh the importance of the clinical implications of our findings. This means that provided the patient is given an antibiotic to which the *E. coli* is susceptible, information on the AMR pattern will not change either the decision management or the advice about symptom development given to the patient. This study is based on two previous studies carried out in the primary care setting. The study of virulence genes is not performed in this setting and has therefore not been studied in this paper.

### 3.3. Relation to the Literature

There is sparse literature in this field. To our knowledge, the study performed by Butler et al. in the United Kingdom is the one most similar to ours [11]. In contrast to our findings, Butler et al. found that the symptoms were longer in duration for patients with UTI caused by *E. coli* resistant to trimethoprim, who were treated with an antibiotic to which the *E. coli* was susceptible. Trimethoprim is one of the first-line antibiotics for UTI in the United Kingdom [12]; hence, the decision to prescribe another antibiotic might have been taken based on unregistered reasons. Consequently, Butler et al. argue that due to unidentified causes, these patients might have been different from the remaining study population.

It is worth noting that the study by Butler et al. is different from ours in several ways. Firstly, they had a broader population as they also included men. Secondly, the study design was different, with patients being interviewed to assess symptom duration one month after consultation and this may have caused recall bias. Finally, resistance rates and first-line choices of antibiotics differ between the United Kingdom and the Nordic countries. Pivmecillinam is a first-line antibiotic in Denmark [14], Sweden [15], Norway [16], and Finland [17]. Furthermore, the resistance rates to pivmecillinam are very low (i.e., between 3 and 6%) [9,18,19,20]. Due to this low resistance rate to the first-line antibiotic, we were not able to assess the influence of resistance to the most commonly used antibiotic, as Butler et al. did.

The functional mechanisms of the first-line antibiotics differ as well. Trimethoprim works as a bacteriostatic drug inhibiting the synthesis of one of the important nucleotides in the synthesis of DNA, whereas pivmecillinam (a pro-drug to mecillinam) works as a bactericide targeting the biosynthesis of the bacteria cell wall, and is a very effective drug against Gram-negative bacteria [21].

Another two studies from the United Kingdom investigated the influence of treatment with an antibiotic to which the strain is resistant on the duration of symptoms of uUTI [10,13]. Although the aims are not fully comparable to that of our analysis, the design can be compared, as both studies used symptom diaries. Both Little and McNulty et al. assessed the duration of moderate to severe symptoms. Little et al. also assessed an overall severity score of symptoms at days two to four [13]. In our analysis, we investigated the course of symptoms in different ways: the daily severity of symptoms (frequency and dysuria) and number of days until the symptoms had disappeared, regardless of severity. For future research, it would be relevant to unify scales to compare data across studies.

A recent study from Iran investigated the importance of virulence genes profiling in uropathogenic *E.coli* and found that one gene, *cnf1*, was present in all UTIs. Our findings relate to primary care where genetic information is not available at this point. Furthermore, for women with uncomplicated cystitis caused by *E. coli* we found that the resistance pattern did not affect the course of symptoms. Therefore, for this group genetic profiling does not seem relevant at the moment [22].

Finally, our findings are in line with those from a study investigating the treatment of uncomplicated pyelonephritis. Talan et al. found that all women infected with a strain resistant to trimethoprim–sulfamethizole and treated with ciprofloxacin were cured 11 days after the initiation of treatment [23]. This is in line with our finding regarding the irrelevance of the AMR pattern provided that the patient is given an antibiotic to which the *E. coli* is susceptible. Nonetheless, it is important to highlight that the use of a broad-spectrum antibiotic (e.g., ciprofloxacin) is not the solution to the problem. This was shown in a recent literature review assessing the efficacy and safety of levofloxacin (a broad-spectrum antibiotic) for complicated UTIs and pyelonephritis [6]. The use of a narrow-spectrum antibiotic (e.g., pivmecillinam) as the first-choice antibiotic is a good option provided there is a low prevalence of resistance.

### 3.4. Implications

The AMR pattern is not clinically relevant in women with uncomplicated cystitis caused by *E. coli* and treated with an antibiotic to which the strain is susceptible. Our results apply for women with uncomplicated cystitis caused by *E. coli*, which is the most frequent cause of uncomplicated cystitis, representing about 80% of all cases [7]. However, we would expect our results to apply for other uropathogens as well, as there is no conclusive evidence that the type of uropathogen influences the course of symptoms for uUTI [10,24,25].

The resistance patterns of *E. coli* strains causing uncomplicated cystitis vary considerably between regions and countries; therefore, specific treatment recommendations may not be universally suitable for all regions or countries. Hence, up-to-date local surveillance data on *E. coli* resistance rates at the community level are crucial to optimize the prescription of antibiotics in primary care. A recent Irish study carried out in the Cork area found a high prevalence of resistance towards several of the antibiotics recommended as empirical treatment for UTI in their national guidelines [26]. Updated guidelines are especially important in settings in which empiric prescription is the only option for the management of patients with suspected uncomplicated cystitis due to the lack of access to point-of-care tests (POCTs). Regarding settings such as Denmark, which have a wide availability of POCTs, urine culture is routinely performed in practice [27]. Hence, for women with uncomplicated cystitis caused by *E. coli*, testing for resistance in this setting is not necessary for GPs to make an appropriate prescription of pivmecillinam, due to the low resistance rates. However, in the Danish setting, the use of sulfamethizole and trimethoprim as first choices for uncomplicated cystitis has been rightfully revised considering the high resistance rates (27% for sulfamethizole and 21% for trimethoprim) [9]. The current first-line treatment for uncomplicated cystitis in Denmark is pivmecillinam, which is also one of the recommendations for first-line treatment according to the newest guidelines from the European Association of Urology [14,28].

## 4. Materials and Methods

This analysis was based on data from two studies on the diagnosis and management of UTI in general practice performed in the Capital Region of Denmark between 2014 and 2016 [27,29]. The inclusion and exclusion criteria of these studies can be found in the previously published protocols [30,31].

For this analysis, we included women ≥ 18 years with symptoms of uncomplicated cystitis (dysuria, frequency or urgency) and a positive urine culture with significant growth of *E. coli* (≥10^3^ CFU/mL) [32]. In Denmark, four different antibiotic regimes were recommended for uncomplicated cystitis at the time the data were collected: 400 mg of pivmecillinam three times a day for three days, 1 g of sulfamethizole twice a day for three days, 200 mg of trimethoprim twice a day for three days and 50 mg of nitrofurantoin four times a day for three days [8]. We analyzed patients who initiated antibiotic treatment with one of the recommended antibiotics one day after the initial consultation to obtain a homogenous sample of patients in terms of recovery time and since this was the most common day for initiating antibiotics in our two study samples.

### 4.1. Antimicrobial Resistance

At the initial consultation, the patient was instructed to deliver a mid-stream urine sample, which was sent to the microbiological laboratory for culture and susceptibility testing.

The urine was sent in tubes containing boric acid to stabilize the bacterial count. The culture was performed on blood agar and CHROMagar or “blue” agar plates. Samples were incubated aerobically over night at 35 °C.

All samples were quantified. *E. coli* was specified based on colonial morphology. Significant growth was defined as ≥10^3^ colony forming units per milliliter (cfu/mL) for *E. coli.* [27,29]. Antimicrobial susceptibility testing was performed with discs containing antibiotics on Mueller–Hinton agar plates according to EUCAST standards (version 3.1, year 2013) [33]. Testing was performed for mecillinam, trimethoprim, nitrofurantoin, sulfamethizole, ampicillin and ciprofloxacin.

### 4.2. Outcome and Covariates

The primary outcome was the symptom severity score measured daily during one week after the index consultation. In this analysis, we focused on the most common symptoms of uncomplicated cystitis: frequency and dysuria [34]. We used the Holm–Cordoba UTI Score which is a validated UTI-specific patient-reported outcome measure (PROM) [35]. According to the Holm-Cordoba UTI Score, frequency is an aggregated symptom complex, based on four specific symptoms (increased daytime frequency, increased urge, increased hurry to visit the toilet, increased incontinence), measured on a scale from 0 to 12. Correspondingly, dysuria is an aggregated symptom complex, based on three specific symptoms (pain by urination, difficulty emptying the bladder, suprapubic tension), measured on a scale from 0 to 9.

The secondary outcomes were (a) the number of days until all frequency symptoms had disappeared, (b) the number of days until all dysuria symptoms had disappeared, and (c) the number of days until all UTI symptoms had disappeared (the latter was assessed through a global item, which was the overall assessment of the symptom burden by the patient), hereafter referred to as the ‘overall cure’.

The exposure was the antimicrobial resistance pattern. In this analysis, we only investigated the influence of antibiotics with a known resistance rate of more than 10%, i.e., ampicillin, trimethoprim and sulfamethizole. We carried out two analyses. First, we examined the influence of an overall AMR pattern with the following three groups: (1) resistant to all three antibiotics (i.e., ampicillin, trimethoprim and sulfamethizole); (2) resistant to one or two of these antibiotics; and (3) sensitive to all three antibiotics. Secondly, we investigated the influence of resistance toward the specific antibiotic: (1) resistance to ampicillin compared to sensitivity to all the three antibiotics; (2) resistance to sulfamethizole compared to sensitivity to the three antibiotics; and (3) resistance to trimethoprim compared to sensitivity to all three antibiotics.

We did not control for confounders, since we had a very limited number of relevant co-variates available in both datasets.

### 4.3. Statistical Analysis

The association of the AMR patterns with the course of the symptom severity one week after the index consultation was assessed in linear mixed models including a subject random effect to account for repeated measurement. The association of the AMR patterns with the number of days until symptoms had disappeared was also assessed in linear models.

In all the statistical analyses, missing information on resistance for the three antibiotics was imputed and the results from multiple (25) imputations were aggregated using Rubin’s rule. Imputation was carried out with chained equations in augmented data on the 356 patients with UTI caused by *E. coli*. Variables used for the imputation with chained equations were as follows: resistance to ampicillin, sulfamethizole, trimethoprim, mecillinam and ciprofloxacin, and sex, (un)complicated UTI and pyelonephritis, clinic area and age. Analyses were carried out with SAS version 9.4.

## Figures and Tables

**Figure 1 antibiotics-11-00188-f001:**
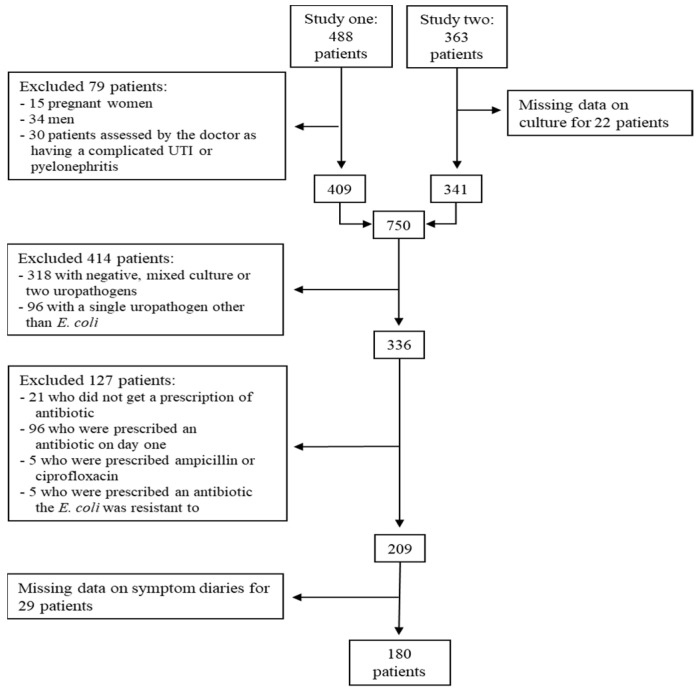
Included patients.

**Figure 2 antibiotics-11-00188-f002:**
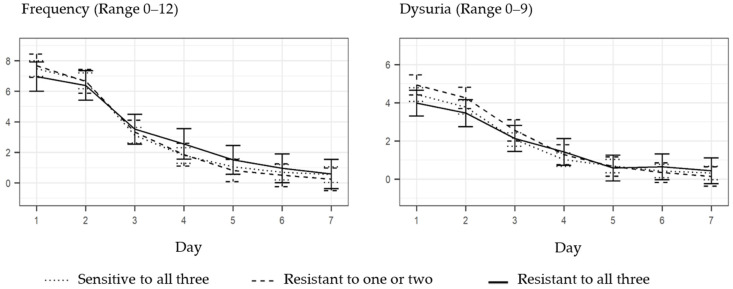
Influence of the overall AMR pattern on daily mean symptom severity scores for frequency and dysuria, with 95% confidence intervals for each day.

**Figure 3 antibiotics-11-00188-f003:**
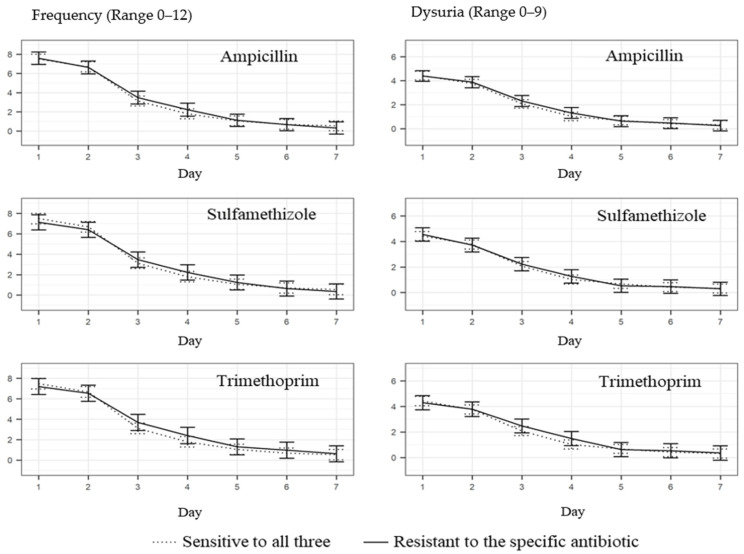
Influence of the AMR pattern on daily mean symptom severity scores for frequency and dysuria. Data are presented per antibiotic with 95% confidence intervals for each day.

**Table 1 antibiotics-11-00188-t001:** Baseline characteristics.

		*n* (%)
Age		
	<65 years	121 (67.22)
	≥65 years	59 (32.78)
Antibiotic prescribed		
	Pivmecillinam	114 (63.33)
	Sulfamethizole	32 (17.78)
	Trimethoprim	11 (6.11)
	Nitrofurantoin	23 (12.78)
AMR pattern (missing data *n* = 42, imputated)		
	Sensitive to all three	103 (57.22)
	Resistant to one or two	48 (26.67)
	Resistant to all three	29 (16.11)

**Table 2 antibiotics-11-00188-t002:** Influence of the antimicrobial resistance pattern on the number of days until symptoms had disappeared.

	Number of Days Until Symptoms Had Disappeared					
	Overall Cure *		Frequency		Dysuria	
	Mean (95% CI)	Days	Mean (95% CI)	Days	Mean (95% CI)	Days
AMR pattern overall						
Sensitive to all three	4.2 (3.9–4.5)	Ref	3.8 (3.4–4.1)	Ref	3.6 (3.2–3.9)	Ref
Resistant to one or two	4.5 (4.0–5.0)	+0.3	3.6 (3.1–4.1)	−0.2	4.1 (3.5–4.7)	+0.5
Resistant to all three	4.7 (4.1–5.3)	+0.5	4.1 (3.5–4.8)	+0.3	3.9 (3.2–4.7)	+0.4
AMR pattern focusing on specific antibiotics						
Resistant to ampicillin	4.5 (4.1–5.0)	+0.3	3.8 (3.3–4.2)	0	3.9 (3.4–4.4)	+0.3
Resistant to sulfamethizole	4.6 (4.1–5.1)	+0.4	3.8 (3.3–4.3)	0	4.0 (3.4–4.6)	+0.4
Resistant to trimethoprim	4.7 (4.2–5.2)	+0.5	4.1 (3.6–4.7)	+0.4	4.0 (3.4–4.6)	+0.5

Antimicrobial resistance to ampicillin, sulfamethizole and trimethoprim. Resistant groups are compared to ‘sensitive to all three’ in linear models. All analyses have *p*-values > 0.05. * Overall cure: Patients felt all UTI symptoms had disappeared.

## Data Availability

Patient-level data in an anonymized format are available on reasonable request to the primary author.

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
