# Peer review of "Influence of Antimicrobial Resistance on the Course of Symptoms in Female Patients Treated for Uncomplicated Cystitis Caused by Escherichia coli"

_antibiotics, 2022, doi:10.3390/antibiotics11020188_

Round 1

Reviewer 1 Report

Some aspects seem to be unclear stated.

  1. There is not stated what type of uncomplicated UTI refers to. From the analyzed symptoms, the present paper seems to refer particularly to cystitis in women, not to uncomplicated UTIs in general which includes also pyelonephritis. 
  2. In the material and methods section, there should be explained more clearly compared groups. 
  3. It can be also explained what guideline recommendations were used for antibiotic prescription. Were there also consulted the European Association of Urology updated guidelines? Are there some local studies about local antibiotic resistance patterns that influence antibiotic administration? 

Author Response

Dear reviewer

Thank you very much for taking your time to review our manuscript and for all your comments.

We have addressed your concerns below and changed the manuscript accordingly.

  1. In the paper we focus on uncomplicated cystitis. We have changed the words in the title and throughout the full article to clarify this.
  2. We have re-written the paragraph about the different groups in the “Material and Methods” section to make it clearer.
  3. At the time when the study took place, there were different guidelines for treatment of uncomplicated cystitis in Denmark. This is already included in the “Material and Methods” section and has been slightly expanded. We have included a reference on the European Association of Urology updated guidelines in the “Discussion” section under “Implications”.

Reviewer 2 Report

This is an unusual study in my opinion, where authors analyzed results from previous studies. I would not find this as a problem if these studies were not starting (according to clinicaltrials.gov) in the year 2015. Therefore, I do not believe this manuscript could conclude on antibiotic resistance in UTI as this could have changed in the last 6 years.

The manuscript itself also has several flaws; the introduction does not have a flow, the sentence should never start with a number, the limitation section is not discussed, etc

Author Response

Dear reviewer

Thank you very much for taking your time to review our manuscript and for all your comments.

We have addressed your concerns below and changed the manuscript accordingly.

  1. The research question focusses on the clinical implications of AMR on the course of symptoms. We found that “Strains of  E. coli resistant to ampicillin, sulfamethizole and trimethoprim causing uncomplicated cystitis do not result in more severe symptoms or longer symptom duration if treated with an antibiotic to which they are susceptible”. These findings are clinically relevant independently of the year for data collection, as changes in the prevalence of AMR would not affect the results. Furthermore, resistance rates have only slightly changed over the last decade. Resistance rates for pivmecillinam have changed from 5% in 2016 to 6% in 2020. For ampicillin, the resistance rates are also similar, with 38% to 36% of the strains being resistant, respectively, and the resistance rates of E. coli to trimethoprim have slightly decreased from 29% to 27% according to DANMAP which is the Danish Program for surveillance of antimicrobial consumption and resistance in bacteria from food, animals and humans.
  1. As suggested by the reviewer we have reworded the introduction and the flow is now more natural. Sentences have been rewritten so they do not start with a number. The limitation section has been expanded and more clearly elaborated.

Reviewer 3 Report

The manuscript was focused on the analysis of Escherichia coli strains which exhibited resistance to ampicillin, sulfamethizole and trimethoprim uUTI. They did not cause more severe or longer symptoms once treated with antibiotics to which they were susceptible.

The article is well written, the aim is clear and clearly supported by results. I suggest it for publication in the present dorm. 

Author Response

Dear reviewer, thank you very much for taking your time to review our manuscript and we appreciate your positive comments very much.

Reviewer 4 Report

Dear Author

Thank you very much for your manuscript submission. The design of your work is not strong.

1. Indeed, the microbial virulence genes and factors determine the severity and syndromes in patients with UTIs (treated and untreated patients). That's why, virulence gene profiling in uropathogens is an effective procedure in this regard.

Please read and add the following paper to the References section of the manuscript to have a fruitful revision in your manuscript.

Clinical cases, drug resistance, and virulence genes profiling in Uropathogenic Escherichia coli. J Appl Genet. 2020 May;61(2):265-273. doi: 10.1007/s13353-020-00542-y. Epub 2020 Jan 16. PMID: 31950434.

2. Please mention the functional mechanisms of applied antibiotics in your work.  In this regard, please read and add the following paper to the References section of the manuscript to have a fruitful revision in your manuscript.

Antimicrobial Agents and Urinary Tract Infections. Curr Pharm Des. 2019;25(12):1409-1423. doi: 10.2174/1381612825999190619130216. PMID: 31218955.

3. Your study is just about women; What about men? You have to change the general title of your manuscript. "Influence of antimicrobial resistance on the course of symptoms in female patients treated for urinary tract infection caused by Escherichia coli"

4. Please write the mentioned percentage with two decimals e.g.,: 16.11% not 16%

5. Please mention the year and the version of EUCAST within the text.

6. Please give references to used protocols.

7. Please mention the unit and the Company of the recruited antibiotics

8. Where are the employed molecular tests and techniques?

Author Response

Dear reviewer

Thank you very much for taking your time to review our manuscript and for your comments.

We have addressed your concerns below and changed the manuscript accordingly.

  1. Thank you for the suggested article about virulence gene profiling in uropathogens. This is a very interesting article. Our findings though, are primarily relevant for primary care practitioners and female patients with uncomplicated cystitis. At this point of care, genetic information is not available, so it does not impact the clinical decision. We have included the suggested article in the references and used it in the discussion.
  2. We have summarized the functional mechanisms of the antibiotics considered in our study as requested by the reviewer and we have added the reference.
  3. Thank you for this inquery. As requested, we have changed the title to “Influence of antimicrobial resistance on the course of symptoms in female patients treated for uncomplicated cystitis caused by Escherichia coli” and also added this through the manuscript to clarify that we only investigate data from women.
  4. The percentages have been changed to include two decimals where data was available for this.
  5. We have included the year and version of EUCAST in the text.
  6. Data for our analysis was based on data from two studies taken place between 2014 and 2016 in general practice in the Capital Region of Denmark. The protocols for these two are included in the references in the “Materials and Methods” section. There was no published protocol for the analysis for this article.
  7. We have included the recommended doses for the antibiotics recommended in Denmark at the time when the data was collected in the “Material and Methods” section. We have not included the companies as several companies market the antibiotics.
  8. Thank you very much for your request. The data from this study was collected from general practice. In this setting in Denmark urine cultures with E. coli are diagnosed based on colonial morphology as molecular test and techniques are not considered relevant here. When the urine is collected from the hospitals different molecular tests are performed. We have added the first mentioned to the manuscript and more details.

Reviewer 5 Report

Good study, well designed.

Please cite also in the introduction: Bientinesi R, Murri R, Sacco E. Efficacy and safety of levofloxacin as a treatment for complicated urinary tract infections and pyelonephritis. Expert Opin Pharmacother. 2020 Apr;21(6):637-644

Author Response

Dear reviewer, thank you very much for taking your time to review our manuscript and for your nice comments.

Thank you very much for the suggested article. We have included this in the “Introduction” section as suggested, but also in the “Discussion” when mentioning the use of broad-spectrum antibiotics for uncomplicated UTIs.

Round 2

Reviewer 1 Report

All the suggestions for modifying the present paper were taken into consideration and I recommended it to be accepted in the present form.

Reviewer 2 Report

I believe This article is now appropriate for publication

Reviewer 4 Report

Accept